# Identification of potential riboswitch elements in *Homo sapiens* mRNA 5'UTR sequences using positive-unlabeled machine learning

**William S. Raymond**[1]*, **Jacob DeRoo**[1], **Brian Munsky**[1,2]

**1** School of Biomedical Engineering, Colorado State University, Fort Collins, Colorado, United States of America, **2** Chemical and Biological Engineering, Colorado State University, Fort Collins, Colorado, United States of America

* williamscottraymond@gmail.com

**Data availability statement:** All relevant data for this study are publicly available from the GitHub repository (https://github.com/MunskyGroup/human_riboswitch_hits) and

## Abstract

Riboswitches are a class of noncoding RNA structures that interact with target ligands to cause a conformational change that can then execute some regulatory purpose within the cell. Riboswitches are ubiquitous and well characterized in bacteria and prokaryotes, with additional examples also being found in fungi, plants, and yeast. To date, no purely RNA-small molecule riboswitch has been discovered in *Homo Sapiens*. Several analogous riboswitch-like mechanisms have been described within the *H. Sapiens* translatome within the past decade, prompting the question: Is there a *H. Sapiens* riboswitch dependent on only small molecule ligands? In this work, we set out to train positive unlabeled machine learning classifiers on known riboswitch sequences and apply the classifiers to *H. Sapiens* mRNA 5'UTR sequences found in the 5'UTR database, UTRdb, in the hope of identifying a set of mRNAs to investigate for riboswitch functionality. 67,683 riboswitch sequences were obtained from RNAcentral and sorted for ligand type and used as positive examples and 48,031 5'UTR sequences were used as unlabeled, unknown examples. Positive examples were sorted by ligand, and 20 positive-unlabeled classifiers were trained on sequence and secondary structure features while withholding one or two ligand classes. Cross validation was then performed on the withheld ligand sets to obtain a validation accuracy range of 75%-99%. The joint sets of 5'UTRs identified as potential riboswitches by the 20 classifiers were then analyzed. 1533 sequences were identified as a riboswitch by one or more classifier(s) and 436 of the *H. Sapiens* 5'UTRs were labeled as harboring potential riboswitch elements by all 20 classifiers. These 436 sequences were mapped back to the most similar riboswitches within the positive data and examined. An online database of identified and ranked 5'UTRs, their features, and their most similar matches to known riboswitches, is provided to guide future experimental efforts to identify *H. Sapiens* riboswitches.

from the figshare repository (https: //doi.org/10.6084/m9.figshare.24587334.v2).

**Funding:** This study was financially supported by the National Science Foundation (NSF) in the form of a grant (1941870) received by WR and BM. This study was also financially supported by the National Institutes of Health (NIH) in the form of a grant (R35GM124747) received by WR and BM, and a grant (1R01AI168459-01A1) received by JR. The funders had no role in study design, data collection and analysis, decision to publish, or preparation of the manuscript.

**Competing interests:** The authors have declared that no competing interests exist.

## Introduction

A riboswitch (RS) is a non-coding RNA sequence harboring a structure with two distinct conformations. Conformational changes are induced when an aptamer region interacts with a target small molecule, revealing or occluding functional parts of the RNA. This inducible structural change allows for broad, responsive regulation of various cellular processes via modification of protein production — achieved by affecting transcription termination/continuation, translation inhibition/activation, mRNA splicing, or mRNA stability [1–4]. Whether a particular riboswitch acts in a positive or negative regulatory manner in the presence of its ligand strongly depends on the expression platform and aptamer location in relation to other elements, such as the ribosomal binding site. These regulatory effects could conceivably have disease-related implications due to under- or over-expression of a regulatory biomolecule or loss of function due a genetic mutation of the riboswitch in question. Besides being a critical regulatory tool in nature, both natural and synthetic riboswitches have been used in synthetic biology for responsive translational control, splicing control, and photo-regulation [5–7]. The current set of described riboswitches is predicted to be a small subset of all existing riboswitches – leading to open questions such as "which riboswitch classes are uncharacterized?" or "why do some life-essential molecules lack known riboswitch aptamers?" Riboswitch discovery has been an active area of research since their first description in 2002, with many computational and experimental efforts undertaken to elucidate new riboswitch classes [8]. Riboswitches occur ubiquitously in prokaryotes, where they enjoy a rich diversity of around 40 known molecular targets [3]. In contrast, nearly every example of a eukaryotic riboswitches in the current literature was found in lower eukaryotes, such as mold, yeast, and fungi, and they leverage thiamine pyrophosphate (TPP) as their target ligand [2,8,9]. Among higher eukaryotes, several species of plants have a single TPP riboswitch in the 3'UTR of the conserved gene THIC – the riboswitch acts to regulate gene expression by creating an unstable mRNA product in the presence of TPP [10]. Interestingly, multiple analogous "pseudo-riboswitches" (riboswitch like elements that are stabilized by proteins and small molecules) have been located within the untranslated region (UTR) of human (*Homo Sapiens*, *H. Sapiens*) translatome [11,12]. The existence of some analogous mechanisms in *H. Sapiens* and a single well-described eukaryotic riboswitch prompts the question: "do riboswitches have an unknown niche in all higher eukaryotes, or are they simply missing?" Indeed, this question is one of the largest open questions in the field today [8]. From a disease perspective, do riboswitches exist within in *H. Sapiens*, and if so, where and with what implications on human health?

Machine learning is routinely used in Bioinformatics for a wide range of RNA related tasks, including parsing RNA-seq data, performing RNA secondary structure prediction, and providing discovery based approaches for RNA sequences, splice sites, and genome wide functional RNA elements [13,14]. For the task of computational riboswitch prediction, previous methods leverage hidden Markov models (HMMER, RiboSW, Riboswitch Scanner [15–17]), covariance models and context free grammar (Infernal [18]), and sequence alignment + computational folding (Riboswitch Finder, RibEx, RNAConSLOpt [19–21]). Other more recent software such as Riboflow utilizes deep learning classifiers such as RNN-LSTM or convolutional neural networks (CNN) for their riboswitch identification [22]. Computational methods available to the public up until 2018 are reviewed in Antunes et al. extensively [23]. Notably, many of these have difficulty extrapolating to unknown riboswitches and rely heavily on a previous knowledge base. Recent breakthroughs have been achieved via reverse homology searching approaches (mutate a sequence without disturbing secondary structure, search for the new mutant in a genome wide fashion), which recently helped to identify a list of

potential purine riboswitches in fungi [24] – however this approach once again suffers from a lack of extrapolation and requires a known starting point structure.

Positive unlabeled learning (PU-learning) is a subclass of binary machine learning classification that attempts to learn from data that only contain positive and unknown, unlabeled examples. In other words, they are used to classify data where the labels of one class are known (label 1) or known and incorrect (labeled 1, truly 0), and the other class are unknown and could either be true examples (label 1), unclassified examples (label 0.5), or negative examples (label 0) [25–28]. Situations producing unlabeled-positive data sets are prevalent in fields such as medical diagnosis (e.g., people with a diagnosis vs. people without a condition vs. people with a condition and no diagnostic confirmation), interest-based applications (e.g., people who engaged with an ad vs. those who did not, since not engaging could be a negative or a neutral reaction), and biology (e.g., a class of known proteins vs. proteins with unclassified but similar function vs. proteins without the same function). PU-learning is routinely applied in molecular biological discovery applications since the advent of big data approaches [29–33]; Proteomics, RNA-seq, or whole genome sequencing quantify virtually all species within a sample whether or not the molecules are characterized, creating a tranche of unlabeled data along with its positive examples [34]. Within the context of RNA, PU-learning has also been used to identify non-coding RNA genes [35], predict circularRNA and piRNA disease associations [36,37], to predict RNA secondary structures [38], and to classify metastasis potential from cancer cell RNA-seq data [39].

In this work, we set out to use PU-learning to identify a group of potential sequences in the *H. Sapiens* mRNA 5'UTR that may contain riboswitches — with the hope of providing a first-pass reduced list for targeted laboratory investigations. 67,683 sequences tagged with "riboswitch" from the non-coding RNA database, RNAcentral, were used as positive examples. 48,031 *H. Sapiens* 5'UTR sequences were obtained from the untranslated region (UTR) database, UTRdb, and used as unlabeled examples. Sequences were sanitized and structural- and sequence-based features were extracted. 20 PU-classifiers were trained on the RS-5'UTR feature sets and validated on single or double holdouts of specific RS ligand classes. The resulting ensemble of classifiers was then examined for the overlap of 5'UTR sequences that were considered as riboswitches (positively labeled). 436 sequences were found to be potential 5'UTR hits across all 20 classifiers. These positively labeled 5'UTR hits were then compared with their most similar sequences within the riboswitch data set via metrics comparing length, dot structure differences, and structural feature similarities, and all results are presented in an interactive display website. GO analysis was also preformed to examine fold enrichment of cellular processes and functions. Further verification of the classifier ensemble was performed by applying the classifier ensemble to a set of 25 synthetic riboswitches, of which 84% were correctly discovered as riboswitches despite having minimal representation of similar synthetic riboswitches in any training data. Using our computationally validated ensemble, we provide a minimal list of *H. Sapiens* 5'UTRs that appear most likely to harbor riboswitch sequences in hopes that these hits could be corroborated with future experimental validation.

## Materials and methods

### 67,683 known riboswitch sequences and 43,081 *H. sapiens* 5'UTRs were collected and sanitized for subsequent PU classification

Two RNA databases were selected for training data: RNAcentral and UTRdb. RNAcentral is a meta collection of many databases of all types of non-coding RNA, and was accessed as the source of riboswitch sequences [40]. JSON information of all entries containing the

tag "riboswitch" were queried from RNAcentral on 8.19.22 and filtered to remove duplicate sequences. To sort the riboswitches by structural class, ligands were parsed from the entry descriptions and any missing ligands were obtained from the corresponding entry's RFAM data [41]. Ligands were further filtered to combine names referring to the same ligand (e.g. "mn", "manganese", "Mn2+" all renamed to "Mn2+"). Cobalamin sub-types such as Adenosylcobalamin were combined under the umbrella of "cobalamin" for ligand labelling. Any protein specific ligand was renamed to "protein" (1 total) and all tRNA ligands were lumped to "tRNA" (3 total). Speculative or synthetic riboswitches (nhA-I motif, duf1646, raiA, synthetic, sul1, blank) were relabeled with 'unknown' as their ligand (1130 total). After ligand relabeling, 73,119 riboswitch sequences remained in the data set. After removing identical sequences, 67,683 penultimate riboswitch sequences were stored for machine learning. Riboswitches targeting cobalamin(s), TPP, S-Adenosyl methionine (SAM), glycine, FMN, purine, lysine, fluoride, and guanidine made up 82% of the riboswitch data set. Other ligand labels such as unknown, molybdenum, GMP, or nickel/cobalt made up less than 2% of the data set each (Fig 1A). A full list of ligands represented in the data set can be found in Table 1.

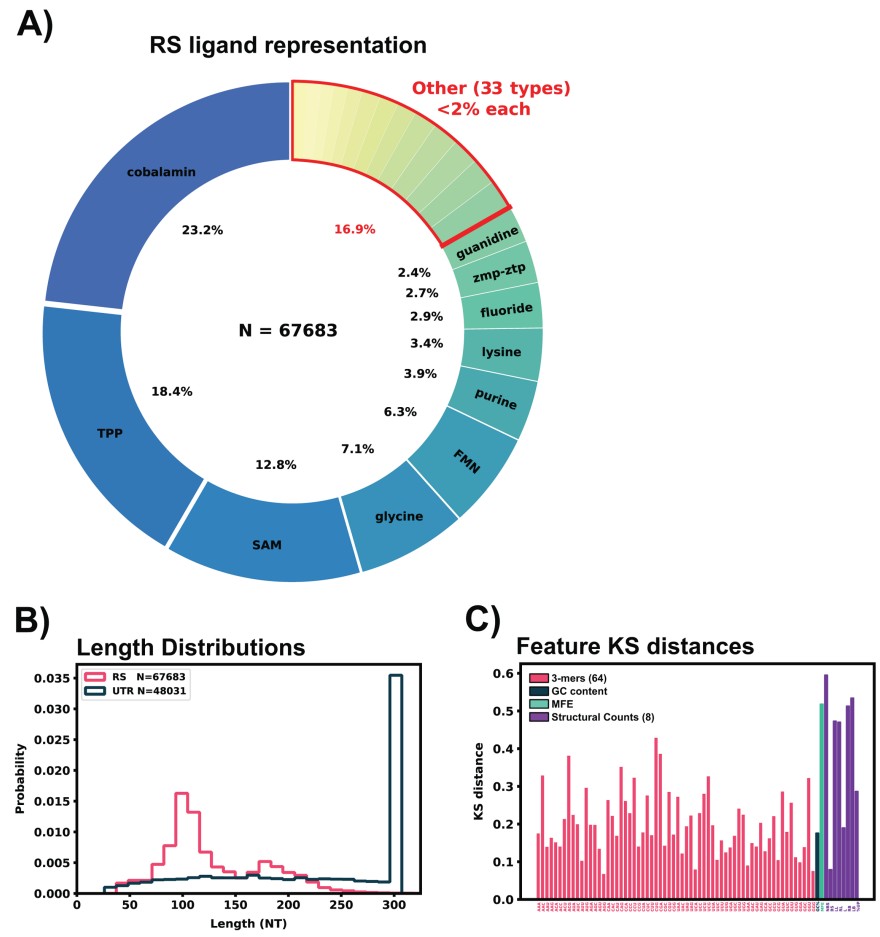

**Fig 1. Riboswitch ligand representation, training data comparisons, and feature extraction of sequence data.** A: Ligand representation within the riboswitch training data (RS). 43 different ligands are represented with 10 ligands having greater than 2% representation in the data set. B: Length distributions for the sanitized 5'UTR and RS data set. C: KS distances between the 5'UTR and RS data set for all extracted features are shown in the bottom panel.

**Table 1. Ligand representation within the data set.**

| Ligand | Count | % | Ligand | Count | % | Ligand | Count | % | Ligand | Count | % |
|---|---|---|---|---|---|---|---|---|---|---|---|
| Cobalamin | 15718 | 23.2 | Molybdenum | 1163 | 1.7 | 2'-dG-II | 30 | 0.044 | Histidine | 1 | 1.5e-5 |
| TPP | 12459 | 18.4 | Unknown | 1130 | 1.7 | aminoglycoside | 21 | 0.031 | Protein | 1 | 1.5e-5 |
| SAM | 8686 | 12.8 | Glucosamine | 1007 | 1.5 | guanine | 20 | 0.029 | Glycine | 1 | 1.5e-5 |
| Glycine | 4835 | 7.1 | Glutamine | 846 | 1.2 | cyclic-di-AMP | 5 | 7.4e-5 | Tetracycline | 1 | 1.5e-5 |
| FMN | 4255 | 6.3 | Mn2+ | 819 | 1.2 | Adenine | 4 | 5.9e-5 | (p)ppGpp | 1 | 1.5e-5 |
| Purine | 2648 | 3.9 | homocysteine | 811 | 1.2 | tRNA | 3 | 4.4e-5 | Alanine | 1 | 1.5e-5 |
| Lysine | 2318 | 3.4 | tetrahydrofolate | 631 | 0.9 | Leucine | 2 | 3.0e-5 | Serine | 1 | 1.5e-5 |
| Fluoride | 1975 | 2.9 | Pre-Q1 | 605 | 0.9 | Tryptophan | 2 | 3.0e-5 | | | |
| zmp-ztp | 1841 | 2.7 | Ni/Co | 569 | 0.8 | Tyrosine | 2 | 3.0e-5 | | | |
| Guanidine | 1640 | 2.4 | GMP | 565 | 0.8 | Proline | 2 | 3.0e-5 | | | |
| Mg2+ | 1308 | 1.9 | cyclic-di-GMP | 526 | 0.8 | Theronine | 2 | 3.0e-5 | | | |
| Methionine | 1175 | 1.7 | glucosamine-6-phosphate | 52 | 0.078 | Valine | 1 | 1.5e-5 | | | |

*H. Sapiens* 5'UTR sequences were pulled from UTRdb, a UTR database of multiple organisims' mRNA [42] during May 2022, and all analyses and computations use this data. UTRdb has since updated the original database to add additional curated functional annotations and UTRs [43]. This update does not substantially affect the sequence data and is not expected to affect any results presented here. For the readers' convenience, we provide the original 5'UTR data from 2022 through the GitHub repository, https://github.com/MunskyGroup/human_riboswitch_hits. Sequences were filtered for identical sequences and stored in a data set. 5'UTR sequences were matched with their corresponding coding regions from the *H. Sapiens* consensus coding region (CCDS) release 22 (accessed 11.28.2021). 5'UTR sequences missing CCDS information were discarded from the data set. 5'UTR sequences were appended with 22 nucleotides downstream from the start codon of the mRNA, and trimmed to the last 300 nucleotides in the 5' to 3' direction if the full 5'UTR sequence was over that limit. This $\min(5'cap, 275\text{NT})$ to start codon to 22 NT region was selected as the area to search for potential riboswitches as regulatory conformational changes in this region could directly block or expose the ribosomal initiation site similar to bacterial riboswitches. After the sanitation, CCDS matching, and length trimming, 48,031 5'UTR sequences were stored for subsequent examination. Fig 1B shows the length distribution of both data sets, and Fig 2A shows an example of a 5'UTR + 25 NT sequence.

For both the known riboswitch data set and the *H. Sapiens* 5'UTR sequences data set, sequences with incomplete or multiple base pair specifications were renamed to the first matching base pair out of the order A, C, G, T/U for the unknown character according to IUPAC naming conventions, Table 2 [44].

## A set of 74 structure and sequence based features were extracted from the 5'UTR and RS data sets

In an effort to collect a broad spectrum of information for machine learning purposes, each sequence was processed to quantify 74 features in two groups: 65 sequence features and 9 predicted structural features.

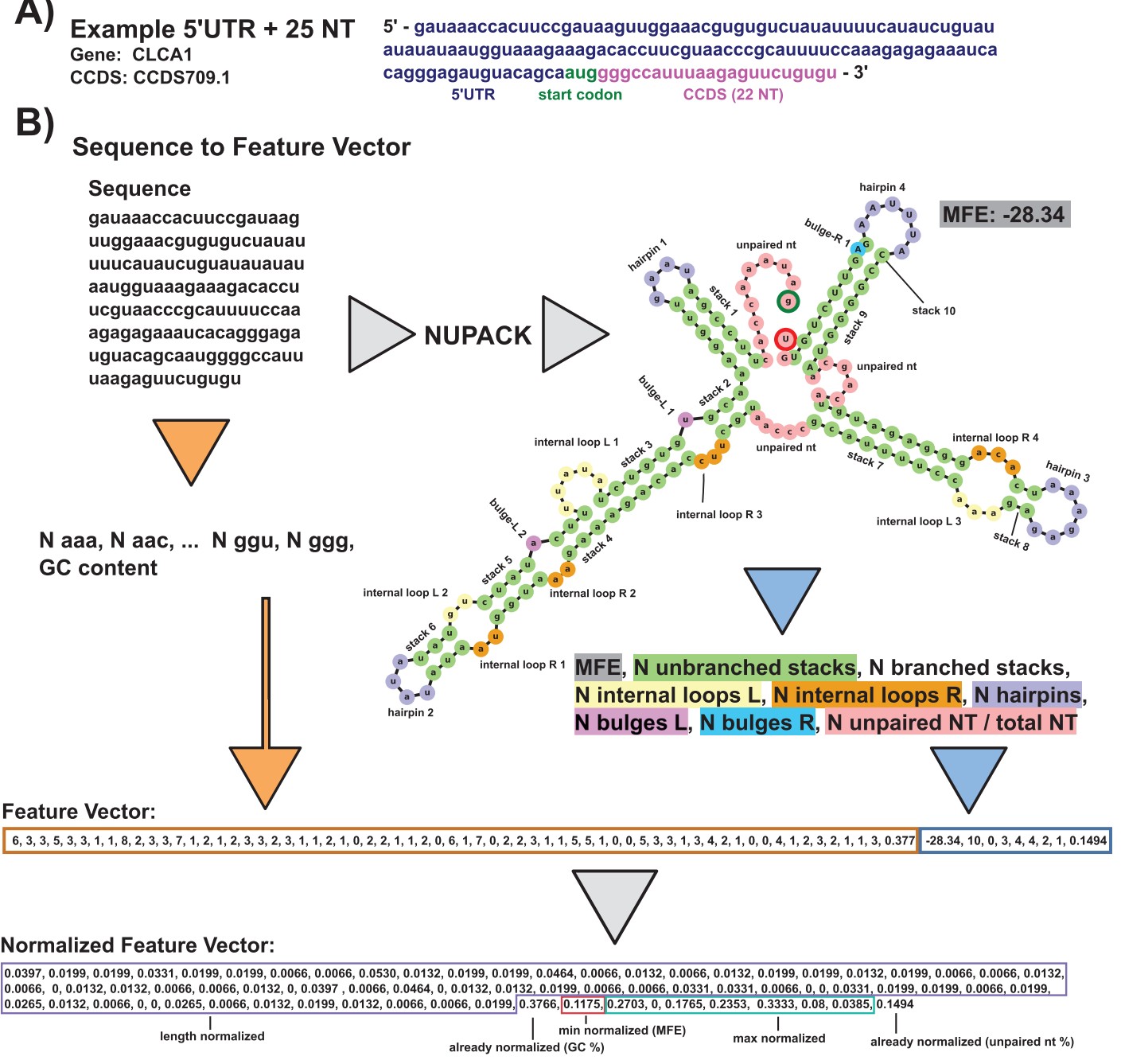

**Fig 2. Example feature extraction of sequence data.** A: Example 5'UTR sequence from the data set containing the start codon and 22 downstream nucleotides (25 total). B: Annotated example of taking an RNA sequence and converting it to a normalized feature vector for our positive-unlabeled learning. For sequence-based features, the sequence is converted into a 3-mer frequency and GC content is calculated. 3-mer frequency is normalized by the number of 3-mer subsets in the sequence (sequence length - 2). Secondary structure based features are generated by passing the sequence through NUPACK. MFE and structural features are extracted from the dot structure. Counts of hairpins, internal loops, bulges, and contiguous stacks (with and without branches) are extracted and max normalized across all the entire data set. Left (L) and right (R) designation corresponds to the 5' to 3' direction and 5' to 3' direction within a base pair stack respectively. MFEs are min-normalized across the data set. The final structural feature considered for learning is the percentage of unpaired nucleotides in the structure. The final output is a vector of length 74 normalized from 0-1.

**Table 2. Nucleotide substitution for data sanitation.**

| IUPAC nucleotide code | Base(s) | Converted to |
|---|---|---|
| A | Adenine | A |
| C | Cytosine | C |
| G | Guanine | G |
| T / U | Thymine or Uracil | U |
| R | A or G | A |
| Y | C or T | C |
| S | G or C | G |
| W | A or T | A |
| K | G or T | G |
| M | A or C | A |
| B | C or G or T | C |
| D | A or G or T | A |
| H | A or C or T | A |
| N | Any | A |

The 65 sequence features include 64 length-normalized 3-mer (AAA, AAG, AAU ... CCC) frequencies, and the GC content. To define these, the $4^k$ sequence $k$-mers were generated for each transcript, and the resulting 64-element vector was normalized by the total number of k-mers (i.e., length - 2) of the corresponding transcript [45,46]:

$$[S_1, \ldots, S_{64}] = \left[ \frac{N_{\text{AAA}}}{L_{\text{seq}} - 2}, \frac{N_{\text{AAC}}}{L_{\text{seq}} - 2}, \frac{N_{\text{AAU}}}{L_{\text{seq}} - 2}, \ldots \frac{N_{\text{GGU}}}{L_{\text{seq}} - 2}, \frac{N_{\text{GGG}}}{L_{\text{seq}} - 2} \right] \tag{1}$$

In addition, the GC content is defined as the count of G and C within the sequence normalized by the sequence length:

$$S_{65} = \frac{N_{\text{G}} + N_{\text{C}}}{L_{\text{seq}}} \tag{2}$$

Structural features were obtained by passing each sequence through the computational folding algorithm, NUPACK 4.0.0.23, [47,48] to obtain a minimum free energy (MFE) secondary structure. NUPACK allows the user to specify a number of RNA strands within one set of "test tube" conditions and provides a list of commonly solved secondary structures and their mean free energies using a computational RNA folding model. Default NUPACK model settings were used when folding all sequences, `Model(material='rna', ensemble='stacking', celsius=37, sodium=1.0, magnesium=0.0)`. For each sequence, the dot structure and the MFE of the most commonly folded non-complexed (no inter RNA strand bonding such as A:A) structure out of 100 RNA strands was saved and recorded as a sequence's secondary structure for feature extraction. A critical limitation of NUPACK is that it does not consider more complex secondary or tertiary structures, such as psuedoknots, and it is possible that future classifiers that have access to more complete structure information could be more predictive. The NUPACK MFE value, unpaired base pair percentage, and counts of how many consecutive stem base pairs in a branching stem or non-branching stem, number of hairpins, number of internal loops left and right, and numbers of left and right bulges were extracted and used as a "structural feature vector" defined as:

$$[S_{66}, \ldots, S_{74}] = \left[ MFE, N_{\text{unbranched stacks}}, N_{\text{branched stacks}}, N_{\text{loops L}}, N_{\text{loops R}}, \ldots. \right.$$
$$\left. N_{\text{hairpins}}, N_{\text{bulges L}}, N_{\text{loops R}}, \frac{N_{\text{unpaired NT}}}{L_{\text{seq}}} \right] \tag{3}$$

Left and right for the bulges and loops were defined as "left" when residing on the 5' to 3' direction of a stack and as "right" when residing on the 3' to 5' direction of a stack. A bulge was defined as a one nucleotide unpaired on either direction interrupting a contiguous stack, loops were defined as two or more unpaired nucleotides interrupting a stack. This naming convention comes from the location when reading the dot structure left to right of the feature: "...((.(.....)))..." has one left bulge and "...((..((.....))..))..." has a left internal loop of two and a right internal loop of three. Unpaired nucleotides are defined as base pairs not within any paired stack, for example, "...(((....)))...(((....)))..." has 9 total unpaired nucleotides as the 8 unpaired nucleotides inside stacks are instead labeled as hairpins. A branching stack is defined as one that has multiple distinct substacks within its stack, e.g. "((...((...))...((...))...))" is one branching stack containing two non-branching stacks. Counts of structural features were max-normalized by the entire combined RS and 5'UTR data set. Fig 2B visually describes the process of taking an example sequence and converting it to its representative feature vector.

A two sample Kolmogorov-Smirnov distance was calculated to compare differences between the known RS features and *H. Sapiens* 5'UTR features. According to the KS distance, most structural features showed a marked disparity between the RS and 5'UTR data set, Fig 1C, while sequence features ranged from 0.05 - 0.4 in their KS distance. Principal component analysis was also performed on the extracted feature set, S1 Fig. This PCA plot shows the overlay of our predicted riboswitch harboring 5'UTRs from the next sections, and demonstrates that PCA alone is insufficient to substantially reduce the number of possible target.

## Results and discussion

### The PU learning model achieves 75% to 99% cross-validation accuracy to identify known but held-out riboswitches

To assess performance of our positive-unlabeled classifiers, we generated 20 separate subsets of training and validation data by withholding specific subsets of the known riboswitches based on their class of ligand. The first ten validation subsets were generated by selecting each of the ten most represented ligand classes (each comprising 2% or more of the overall data) and leaving each one out: Cobalamin, guanidine, TPP, SAM, glycine, FMN, purine, lysine, fluoride, zmp-ztp. The next nine subsets were generated by leaving out pairs of the most commonly represented ligands: FMN+glycine, FMN+SAM, FMN+TPP, FMN+cobalamin, TPP+cobalamin, TPP+glycine, TPP+SAM, cobalamin+SAM, cobalamin+TPP. The final (and most diverse) validation set was created by selecting all riboswitch ligand classes with less than 2% representation in the full RS data set (11305 sequences, 34 ligand classes, 16.9% of the entire RS data set).

Validating each classifier on structural classes that were not provided gives a reasonable confidence that the classifier can extrapolate to riboswitches that are not included in the training set – the target task for eukaryotic riboswitch discovery. 20 Unweighted Elkan & Noto classifiers were trained on the 20 data subsets. Fig 3A shows positive example training accuracy (outer ring), validation accuracy on withheld ligand sets (middle ring) and the positively labeled 5'UTR count (inner ring) of all 20 classifiers. Validation accuracy ranged from 75% to 99% across the classifiers. The classifier validated with withheld TPP riboswitches had high validation accuracy (97%), which is an encouraging sign as all currently described eukaryotic riboswitches use TPP as their ligand [49–54]. The "other" classifier trained the diverse validation set of 34 ligand classes achieved an 89% validation accuracy, demonstrating a surprising ability to extrapolate to examples that are dissimilar from most riboswitches and underrepresented in the training data. To rank the 5'UTR hits based on the ensemble probability of the

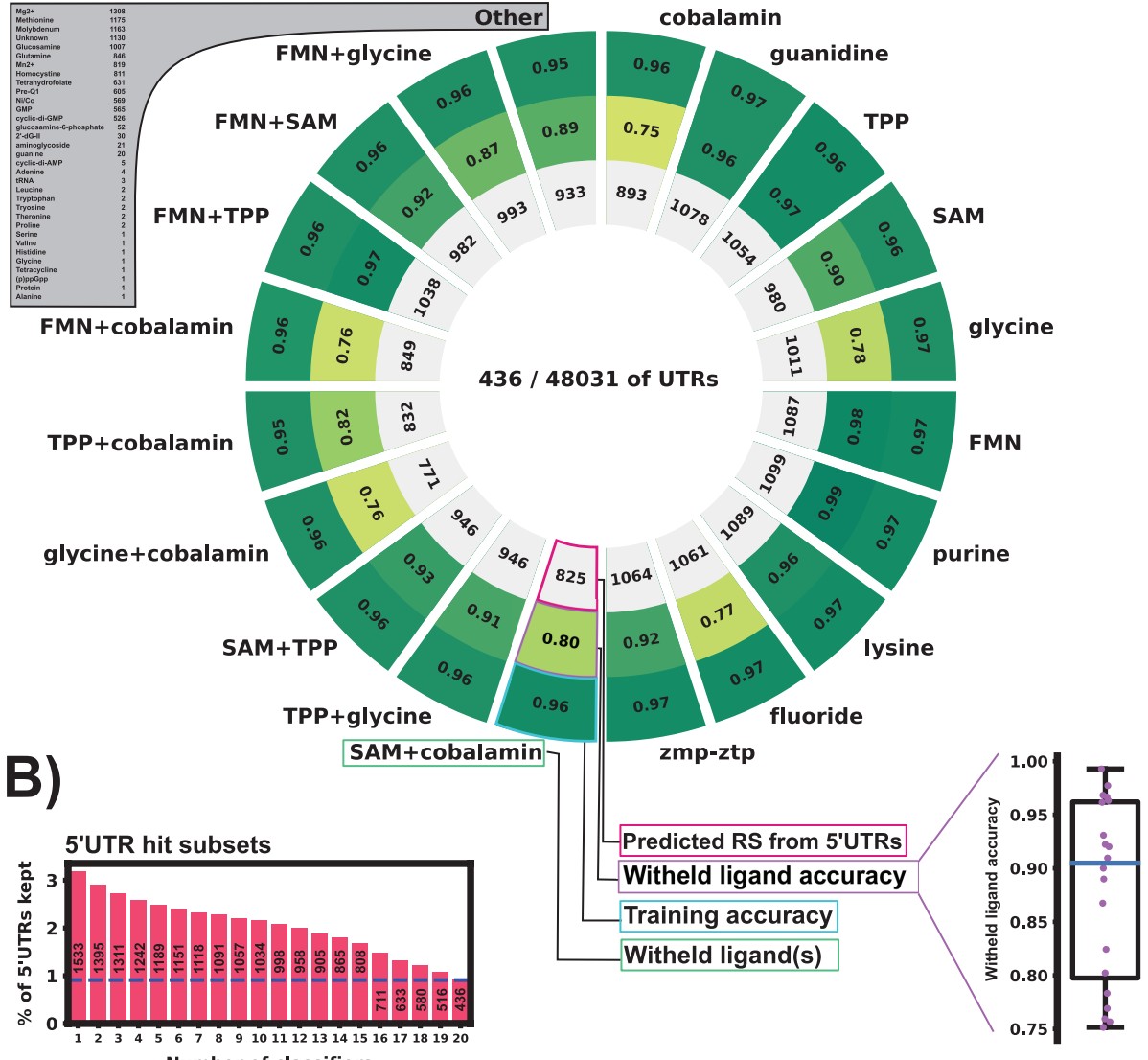

**Fig 3. Training and validation results of 20 PU classifiers.** A: Training and validation results. Each slice represents one PUlearn Elkanoto-classifier trained on a data set withholding one or two ligand-specific riboswitches. The outer ring shows the training accuracy on only positive examples (RS). The middle ring is the validation accuracy on the withheld riboswitch(es) of a particular ligand(s) class. The inner ring shows number of the predicted positive labeled 5'UTR sequences out of the 48,031 5'UTR sequences. The sub-panel on the bottom right shows the withheld validation accuracy (rounded to 2 digits) in a box plot. 436 5'UTRs were selected by all 20 classifiers as positive labeled – potentially harboring riboswitch-like features. B: 5'UTR hit subsets detected by varying numbers of classifiers (1 – 20, full sequences).

classifiers, we average the normalized outputs of the 20 PU classifiers to compute an ensemble probability score from 0 to 1, $J_{\text{Ensemble}}$:

$$J_{\text{Ensemble}}(UTR) = \frac{1}{20}\sum_{i}^{20}\frac{PU_i(UTR)}{\max(PU_i(All\ RS))} \tag{4}$$

where $PU_i(sequence)$ is the positive-unlabeled classifier with the $i$th withheld ligand set, which is normalized by its maximum output over all true RS.

After training and testing the ensemble, feature importance was calculated using sklearn's `permute_importance` function. For each classifier, each of the 74 features was individually scrambled in 10 runs and accuracy loss was assessed on 5000 5'UTR and 5000 RS examples, S2 Fig. Structural features were found to be the most informative and crucial to retain the ensemble accuracy, with unbranched stack counts (UBS) and mean free energy (MFE) being the two most important features across all classifiers. With the exception of GC% and select nucleotide triplets (CCG, CGA, CUU, and GCG), most individual sequence features were found to have minimal impact on accuracy for the ensemble classifiers.

## The ensemble PU learning model correctly identifies 84% of previously unseen synthetic theophylline riboswitches

As an additional verification step of our machine learning approach, we applied our ensemble of 20 classifiers to a wholly synthetic riboswitch data set, one that is not represented anywhere within the training set. The training set used to train the ensemble included 14 total synthetic riboswitch sequences, none of which use theophylline as a target ligand. 25 current theophylline riboswitch sequences were obtained from Wang et al. [55] and were passed through our feature extraction and ensemble classification. Our ensemble correctly identified 84% (21 out of 25) with a selection threshold of 0.95 average ensemble output; With a looser selection threshold of greater than 0.5, 92% (23 out of 25) are selected as riboswitches. This finding indicates that our ensemble can correctly extrapolate to the synthetic riboswitches, with a similar accuracy to the cross validation performed above on natural sequences, even on a riboswitch created through SELEX [56].

## The ensemble PU learning model demonstrates high accuracy to identify known non-human eukaryotic riboswitches

Within the riboswitch data set, there are 884 sequences that came with species information indicating a eukaryotic origin. Of those sequences, 831 appear to be valid sequences (TPP Ligand switches with no quality control flag for their corresponding entry on RNAcentral). Since our objective is to find new riboswitches with unknown ligands, we reasoned that it would illustrative to examine how well our ensemble would perform on these 831 sequences, especially the particular classifiers that are excluding TPP ligand switches from their training data. Classifiers without TPP ligand data correctly selected 85.4%, 86.0%, 82.8%, 81.2%, and 85.1% for classifiers that withheld TPP, TPP and glycine, TPP and cobalamin, and TPP and FMN, respectively. This success rate is comparable to the ensemble classifier that had access to TPP switches, which correctly identified an average 88.5% +- 2.8% of the eukaryotic riboswitches within the data set across all classifiers.

## The ensemble PU model robustly rejects sequences expected to harbor no riboswitches

Calculating the false positive rate for the PU classifier ensemble would require experimental analysis to test each 5'UTR selected by the classifier and is beyond the scope of this manuscript. Following the logic of Elkan and Noto [27], the ranked list of 5'UTR hits should represent those whose features are most similar to features extracted from known riboswitches. Therefore, as a negative control, we test our ensemble model by including sequences that are confidently expected not to harbor riboswitches. For sources of likely

## Ensemble results when training with Exon and Random seqeuences

| | Riboswitch | 5'UTR | Exon | Random | |
|---|---|---|---|---|---|
| N total | 67683 | 48031 | 5946 | 60000 | |
| average ensemble % labeled RS | 95.033% | 7.081% | 0.146% | 0.019% | P(seq) > 0.5 |
| N found | 64320 | 3400 | 8 | 11 | |
| average ensemble % labeled RS | 82.037% | 2.044% | 0.020% | 0.000% | P(seq) > 0.95 |
| N found | 55525 | 981 | 1 | 0 | |
| | positives | positives + false positives | false positives | false positives | |

**overlap with original ensemble: 392/436**

**Fig 4. Ensemble training results when retrained with length normalized *Homo sapiens* exon sequences and random nucleotide sequences.** The blue highlighted box represents the sequences labeled as riboswitches with an output threshold of 0.5, the red box displays a stricter threshold of 0.95. Percentages reported are the average percentage across the 20 classifiers inside the ensemble.

negative sequences, we generated 60,000 random nucleotide sequences. We also collected a random set of 5946 *Homo Sapiens* exon sequences from Ensemble (Ensemble genes 112, GRCh38.p14) to provide more structured, non-random negative sequences. Both sets of these switch-negative sequences were length normalized to the riboswitch data set (Fig 1B), and their features were extracted as described above. We then trained a new 20-classifier ensemble including the riboswitch (label 1), 5'UTR (label 0), exon (label 0) and random sequences (label 0). Average ensemble false positive rates across the 20 classifiers are shown in Fig 4; a breakdown of the False Positive Rate (FPR) by classifier is shown in S3 Fig. Our feature selection and ensemble approach leads to a overwhelming rejection of the presumed negative sequences (1/5,946 exons and 0/60,000 random sequences) while keeping a much larger subset (981/48,031) of 5'UTRs. While this does not confirm the existence of riboswitches within our ensemble selected 5'UTRs, this result shows that our approach successfully disregards the vast majority of negative control sequences. The ensembles with and without the random sequences and exons are highly consistent (392 out of 436 overlap in detected hits). However, for simplicity, the remainder of this paper presents only the results from the original ensemble (i.e., without exons and random sequences).

### PU learning with the trained-ensemble model identifies and ranks a set of 1533 potential 5'UTR riboswitch hits

Now that the ML ensemble has been verified through cross-validation, it is instructive to examine which 5'UTR sequences have been identified as potential riboswitches. Among the classifiers, 436 5'UTRs were identified as harboring potential riboswitch elements by all 20 of the classifiers using a selection criteria of ≥0.95 non-normalized classifier output. Fig 3B shows the relative overlap of all 5'UTR sequences identified by one or more classifier with the same selection threshold. By contrast, the amount of 5'UTR sequences identified as

riboswitches by one or more classifiers was 1533. The existence of an overlap when using all 20 classifiers instills confidence in our ensemble approach. If there was a precipitous drop in identified sequences when using more and more classifiers, that would imply that classifiers are individually identifying completely different subsets of the 5'UTR data set to consider as riboswitches. A drop from 1533 hits to 436 hits when increasing the amount of classifier agreement is substantial, but still leaves a large overlap found by all 20 trained classifiers.

## The classification of top 5'UTR hits is robust to truncation of their 5' ends

Some potential 5'UTR hits may be discarded in our analysis due to the choice to use the full 5'UTR within the classifier. Because many 5'UTRs can be large with multiple regulatory elements, the full 5'UTR could obscure small riboswitch elements that are located near to the start codon. To evaluate how many potential hits may be lost by ignoring sub-sequences of the 5'UTR during classification, we took our 5'UTR data set and for each sequence, we generated 20 sub-sequences for each 5'UTR by truncating the mRNA at 20 evenly spaced locations upstream of the start codon, starting 30 NT from 5' end. For each sub-sequence, we extracted the new features and applied the ensemble classifier. Fig 5A shows the resulting probability of selection as a riboswitch versus the fraction of the sequence used for all (48,031) 5'UTRs; Fig 5B shows the same result but only for the subset of 436 5'UTRs (0.91%) that were previously identified as likely riboswitches using the full length sequence; and Fig 5C shows the same result but for a distinct subset of 1210 5'UTRs (2.5%) that would have been identified as a riboswitch by five or more partial-length sub-sequences, but *not* using the full sequence. Although the probability that a given sequence being a riboswitch increases when using sub-sequences, the vast majority of 5'UTRs (97%) are still discarded as unlikely to be riboswitches. Moreover, for the 5'UTRs that were identified as a riboswitch using their full sequences (n = 436), nearly half (45.5%) of these 5'UTRs are still detected as a riboswitch even when 85% of their sequence is discarded. For example, AUH is consistently detected as a riboswitch with ≥95% confidence for nearly every sub-sequence Conversely, a small fraction of 5'UTRs, such as ATF1, is only identified as a riboswitch when the sequence is 90% or more intact.

From a practical perspective, using the full ensemble and full sequences down-selects to more manageable number of potential hits; Given the ultimate goal of reducing the potential sequence space to an experimentally viable number, 436 is considered to be acceptable amount for future experimental validation. However, the remaining hits from sub-sequences can be revisited and examined as needed and are provided within the supplemental data.

## Identified 5'UTR hits share remarkable feature similarities to known riboswitches

Now that our ensemble classifier has identified a subset of 5'UTRs that may harbor potential riboswitches, it is illustrative to examine the properties of these hits. It is infeasible to manually compare all 436 hits to the RS data set, so to aid with comparison, a GitHub page (https://MunskyGroup.github.io/human_riboswitch_hits_gallery/about/) was created to display, rank, and compare each 5'UTR to its most similar matches in the RS data set. The website contains the subset of 5'UTRs identified by all 20 classifiers as potential riboswitches (436), as well as any 5'UTR identified as a riboswitch by any individual classifier (1533).

5'UTR to RS comparisons can be calculated several different ways, but for the purpose of the website, each 5'UTR was compared to each RS with a using a weighted combination of three metrics: sequence length difference, structural feature vector mean-squared difference, and the predicted 5'UTR MFE dot structure to RS dot structure Levenshtein distance (edit

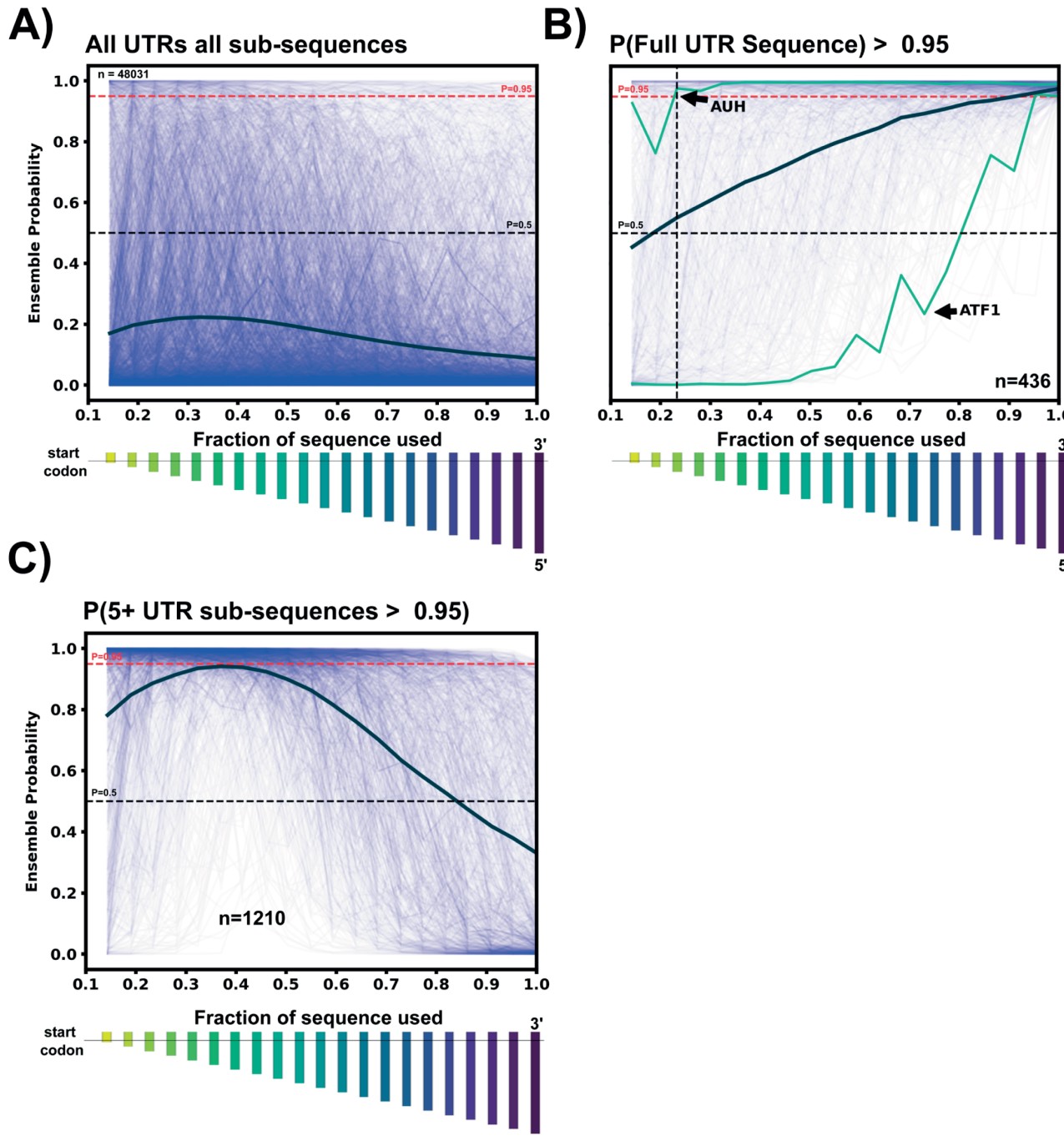

**Fig 5. 5'UTR sub-sequence exploration.** A: For each 5'UTR sequence, 20 evenly-spaced sub-sequences were generated after the first 30 nucleotides in the 3'-5' direction, ensuring the start codon is in all sub-sequences. The relative size of each sub-sequence as a bar chart below the x-axis. For all 5'UTRs in the data set, variable-length sub-sequences were passed through the ensemble classifier to obtain the riboswitch probability. The riboswitch ensemble probability is plotted for each 5'UTR sub-sequence vs. the fraction of the sub-sequence to total 5'UTR length (thin blue lines). The thick dark blue line represents the average ensemble probability for that particular sub-sequence bin. C: Same as A, but only for the 5'UTRs whose full sequences were classified as ≥95% riboswitch by the ensemble. Many 5'UTR sequences such as AUH are classified as a riboswitch until almost 80% of the original sequence is removed. In contrast, some sequences such as ATF1 are no longer considered a riboswitch once 10% of the sequence is removed from the 5' end. Once again the thick dark line represents the average probability of each sub-sequence bin. C: To find sub-sequences not included in the 436 hits, 5'UTR sequences not detected as a riboswitch by the full sequence but were detected as ≥95% riboswitch in 5 or more sub-sequence bins were selected. These 1210 5'UTR sequences and their sub-sequence ensemble probabilities are plotted vs sub-sequence fraction. 1210 sequences could be included as potential riboswitch hits by removing some amount of 5' end nucleotides.

distance). Length distance was calculated as:

$$D_{\text{L}} = (|L_{\text{UTR}} - L_{\text{RS}}|) \tag{5}$$

Likewise, the structural feature metric is also measured by the squared difference between any two extracted feature sets:

$$D_{\text{struct}} = \sum_{66^{th}\text{ feature}}^{74^{th}\text{ feature}} ([S_{5'\text{UTR}}]_i - [S_{\text{RS}}]_i)^2 \tag{6}$$

Finally, the dot structure metric is measured by the Levenshtein distance or "edit" distance between two strings – in other words, how many edits (insertions, deletions, substitutions) to convert one string into another? Eq 7 shows the recursive letter by letter formulation of the Levenshtien distance, where tail(string) refers to everything but the first letter of any given string. If we define $a$ as the UTR sequences and $b$ as the RS sequence, the Levenshtien distance can be computed as:

$$D_{\text{Lev}} = \begin{cases} \text{length}(a) & \text{if length}(b) = 0 \\ \text{length}(b) & \text{if length}(a) = 0 \\ \text{lev}(\text{tail}(a), \text{tail}(b)) & \text{if } a[0] = b[0] \\ 1 + \min \begin{cases} \text{lev}(\text{tail}(a), b) \\ \text{lev}(a, \text{tail}(b)) \\ \text{lev}(\text{tail}(a), \text{tail}(b)) \end{cases} & \text{otherwise} \end{cases} \tag{7}$$

The combined similarity between the UTR and RS is denoted as $J_{\text{Sim}}$ and is computed by normalizing each of the above-described distances by the corresponding maximum value for that metric over all RS in the data set. For length and Levenshtein distances, the maximum distances are: $D_{\text{Lev}} = \max(((476 - 25) - L_{5'\text{UTR}}), L_{5'\text{UTR}})$, and $D_{\text{L}} = 476 - 25$, where 476 is the length of the largest RS sequence in the training data (all UTR sequences are truncated to 300 or less) and 25 is the smallest sequence length within the UTR and RS data set. For the structure feature distance, the normalization factor is obtained by comparing the 5'UTR in question to the entire RS data set and finding the maximum. The combined similarity score is then defined on a scale from 0 (no similarity) to 1 (perfect similarity) according to:

$$J_{\text{Sim}}(UTR, RS) = 1 - \frac{1}{3} \left( \frac{D_{\text{L}}}{\max(D_{\text{L}})} + \frac{D_{\text{Lev}}}{\max(D_{\text{Lev}})} + \frac{D_{\text{struct}}}{\max(D_{\text{struct}})} \right) \tag{8}$$

Because it would be impossible to describe the structure for every selected 5'UTR selected by the PU model, we constructed a display website that tabulates our ensemble results for easy visualization. Each 5'UTR entry is displayed on the website alongside its top three $J_{\text{Sim}}$ matches within the RS data set.

An example website page is presented in Fig 6. The top table displays a 5'UTR and its three most similar (i.e., highest $J_{Sim}$) riboswitches from the training data set. Each column represents a 5'UTR hit or a riboswitch entry from the training data. For each sequence, the predicted MFE secondary structure from NUPACK is displayed for visual comparison. The ligands of the top 20 5'UTR-RS $J_{\text{Sim}}$ matches are also displayed as a preliminary prediction for the potential ligands for that structure. However, future experimental validation would be necessary to ascertain if these hypothetical matches are correct; the potential ligands are

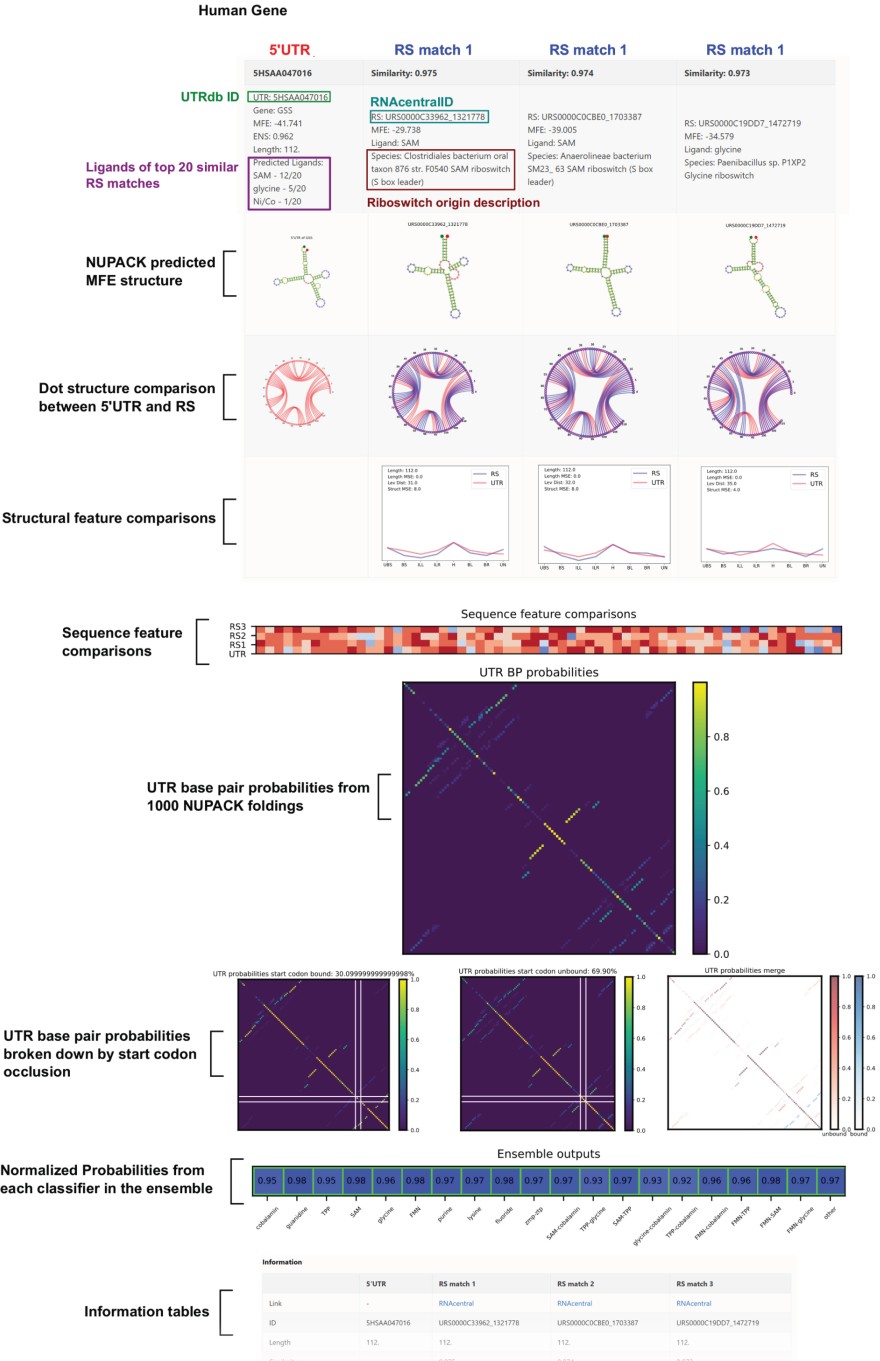

**Fig 6. Example 5'UTR hit display from the website.** The display website (https://MunskyGroup.github.io/human_riboswitch_hits_gallery/_mds/GSS/) provides information on a given 5'UTR detected by the ensemble as a potential riboswitch. Alongside each 5'UTR sequence, information on the top three riboswitch $J_{Sim}$ matches to the 5'UTR are displayed in each column. First row provides information on a given sequence, UTRdb or RS id, source species, and MFE of the predicted structure. The next row displays the NUPACK predicted MFE secondary structure for each sequence. Below that are chord plots representing the bonded base pairs for each RS sequence overlapping the 5'UTR chord plot. The next row shows the normalized structural feature vector comparison for structure counts for the 5'UTR and a given RS. $J_{Sim}$ is reported in these plots. UTR base pair probabilities from 1000 NUPACK foldings of the 5'UTR sequence are shown as a heat map to show multiple potential structures or conformers. Ensemble outputs of each of the 20 classifiers are shown as the last graph before the information tables. Additional information such as the dot structure, origin sequence, and counts of structural features are presented in the information tables below the comparison plots.

presented here more as an indicator of which ligand class from the prokaryotic data is most represented in $J_{Sim}$ matches to a given 5'UTR sequence. Base pair comparisons are also shown for each 5'UTR to RS pair with a circle plot of each RS-5'UTR structure overlayed. Below that is a comparison of each sequence's secondary structure features (stacks, loops, hairpins, etc). The counts of each of these secondary structure features and the sequence dot structures are provided in tables on the bottom page as well. Sequence feature comparisons are shown by a heat map below the structural feature comparison.

Although our ensemble was trained on the highest MFE structures, base pair probabilities from 1000 NUPACK foldings are shown as a probability chart for visualization of conformers (not including pseudoknots). NUPACK foldings were also split into two categories for a more targeted display of potential conformers: Unbound, with the whole start codon available and not in a base pair, or Bound, with any nucleotide of the start codon occluded by a base pair. Base pair probability charts for Bound, Unbound and both merged are shown below the total 1000 folding base pair probability chart. The final plot shows the actual normalized ensemble output, with green boxes highlighting which classifiers considered this 5'UTR as a riboswitch. For the user's convenience, the website's homepage has a table of all webpages that can be sorted by 5'UTR similarity score to known RS ($J_{Sim}$) or by the ensemble probability from the PU classifier ($J_{Ensemble}$, see Eq 4).

## Gene ontology points to enrichment of downstream *H. sapiens* proteins associated with small molecules and transcription/translation regulation

A predominant function of bacterial riboswitches is to regulate the proteins directly related to the riboswitch's target ligand. For example, a fluoride riboswitch may turn on genes useful for processing or mitigating fluoride for an organism [1,4]. With this in mind, it is informative to examine the downstream proteins from the *H. Sapiens* 5'UTR hits for correlations in protein function, looking for genes associated with processing or synthesizing small molecules. Gene ontology (GO) analysis was preformed on the list of 5'UTR hits to look for any cellular function or process enrichment using the PANTHER database [57–59]. GO process results are shown in Fig 7. The process ontology with significant fold enrichment fell into the following categories: Chromatin remodelling, transcription / translation regulation, mRNA splicing, mRNA and rRNA modification, and mitochondrial ubiquinone synthesis. These enrichment results suggest a potential for small molecules to play a regulatory role in gene regulation, even if we cannot comment fully on our 5'UTR hit list without experimental validation. Interestingly, proteins directly involved in chemical stimulus detection were "unenriched" with no proteins found at all. This observation of mutual exclusion between potential riboswitches and sensing proteins is also reasonable - if there are already proteins capable to sense and respond to their intended stimulus, then there is no need to execute redundant functions in riboswitches.

GO function analysis showed a significant enrichment of downstream proteins implicated in binding various small molecules: nucleotides, nucleosides, ubiquinone, and various cyclic compounds, Fig 8. RNA binding and nucleotide binding molecules were extremely enriched with p-values ranging from $10^{-7}$ to $10^{-17}$. Notably there is a negative enrichment of G protein-coupled receptors, this could be explained by riboswitches having direct signalling activity to a cell, bypassing typical trans-membrane signalling pathways.

## Some potential mRNA with potential riboswitches encode proteins with direct involvement in small molecule functions

Although we have no experimental validation for our computationally discovered 5'UTR hits, it is illustrative to highlight and comment several interesting matches found in our

# GO Process fold enrichment

**Fig 7. GO process analysis with ID's and terms.** The left column lists the GO ID and term. Multiple arrows indicate GO term sub-levels. The left bar chart shows fold enrichment for that GO term with significance indicator. The second bar chart shows the log space of P-value significance for each enrichment.

# GO Function fold enrichment

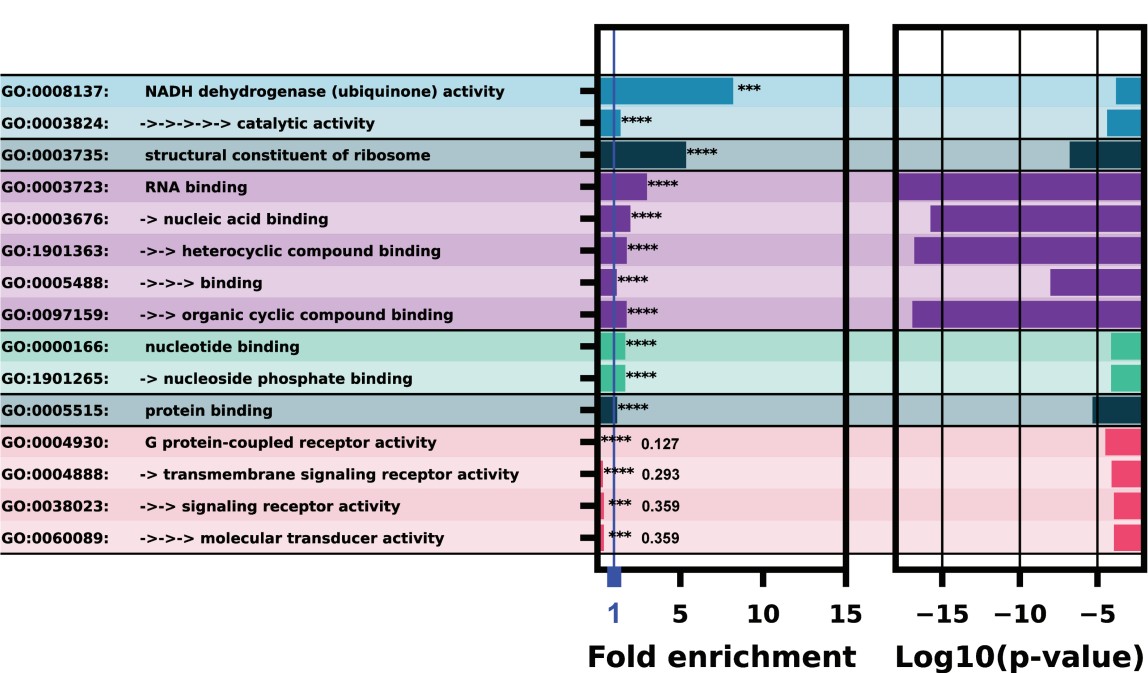

**Fig 8. GO function analysis with ID's and terms.** The left column lists the GO ID and term. Multiple arrows and indents indicate GO term sub-levels. The left bar chart column shows fold enrichment for that GO term with significance indicator. The second bar chart shows the log space of p-value significance for each GO term.

computational analysis. Several computational hits such as AUH and FTSJ1 have roles in processing already known ligands. AUH plays a critical role in leucine degradation, hydrating 3-methylglutaconyl-CoA to 3-hydroxy-3-methyl-glutaryl-CoA [60]. FTSJ1 is known to bind directly to S-adenosylmethionine (SAM) with a potential role in modifying ribosomal RNA and tRNAs [61].

Ubiquinone (CoQ10, CoQ) is an essential antioxidant component of the mitochondria [62, 63]. Many proteins directly related to Ubiquinone synthesis or binding were selected as harboring riboswitches: Biosynthetic proteins COQ3, COQ9, COQ7; Several respiratory complex I proteins: NDUFA2, NDUFA8, NDUFB6, NDUFB9, NDUFS1, and a respiratory complex III protein: UQCRQ. No ubiquinone binding riboswitches are currently described in the literature, although riboswitches binding other critical components of the electron transport chain such as NAD+ have been described [64,65]. Our overrepresentation of riboswitches within the mammalian mitochondria aligns with theories put-forth within Venkata Subbaiah et al. [12], where they speculate that the mitochondria's reduced genome may still harbor RNA switches for mechanism for translational control.

Some proteins represented in the hit list are implicated in small molecule or amino acid synthesis or processing. GSS is responsible for the second step of glutathione synthesis [66], PNPO directly converts vitamin B6 into its active form [67]. Finally, several close matches in predicted structure and feature vectors should also be noted to be of interest: ZNF480, SPAG11B, UBAP2L-0, and TTPAL; However, to our knowledge, these proteins have no clear relation to small molecule processing.

## Conclusion

We have trained an ensemble of machine learning riboswitch classifiers using leave-one-out cross validation of ligand classes consisting of 20 individual classifiers using sequence and predicted RNA structural features. Using this ensemble classifier, we identified a subset of the *H. Sapiens* 5'UTR predicted to harbor riboswitch-like elements (Fig 3). This subset provides a prioritized list of sequences and genes to examine first when designing exploratory experiments. This 436 sequence subset additionally shows positive GO fold enrichment results for downstream genes in many processes with direct small molecule involvement (Figs 7 and 8).

Our approach provides a complementary strategy to the that taken in Mukherjee et al. [24], where the authors began with a known riboswitch sequence and selectively mutated nucleotides while preserving structure, and then searched genomics data for a sequence match. In contrast, our approach starts with genomics data to learn our classifier and any given sequence can be assessed for riboswitch probability. While our approach may be less targeted to the discovery of a riboswitch with a particular structure, it holds the potential to extrapolate beyond single specific structures, as exemplified by its identification of known synthetic riboswitches. Searching for a riboswitch structure in a branch of life vastly different than where riboswitches are previously described likely needs this extrapolation ability.

In future work, our approach could be replicated using the *H. Sapiens* 3'UTR, where described *H. Sapiens* pseudoriboswitches have been found. However, for the purposes of this paper, we have limited our search to the 5'UTR because the bulk of our training data (Bacterial riboswitches) act near their ribosomal binding site, equivalent to the 5'UTR, and looking in the *H. Sapiens* 5'UTR first gives the best chance for efficient machine learning extrapolation. Our approach could also be applied to other eukaryotic UTRs when experimentally validated. In this paper, we also briefly explored the how the detection of potential 5'UTRs riboswitches could be expanded by varying how much of the 5'UTR sub-sequence is used for the identification. This extended list from sub-sequences could could be useful for finding better candidate sequences should the 436 5'UTR hits not bare fruit.

At present we have conducted an entirely computational investigation without experimental validation for any of our final detected hits; Despite this we have labored to provide as much computational validation as possible. We have tested our approaches on sequences expected to harbor no riboswitches as a negative control, ensuring our discovery rate is much higher than the false positive on the negative sequences. We have performed additional promising validation on synthetic riboswitches as well as showing our ensemble correctly labeled known eukaryotic TPP riboswitches. We hope this extended validation as well as our provided data set sorting and display website will prove helpful to the broader scientific community in the experimental search for a riboswitch within the *H. Sapiens* translatome and beyond.

## Supporting information

**S1 Fig. Principal component analysis results of our extracted features do not separate easily.** A) The selected 5'UTR hits (436) heavily overlap with the riboswitch data set principal components. B) Histogram view of the PCA of both data sets along principal component 1.

**S2 Fig. Feature importances averaged across the ensemble indicate that specific nucleotide triplets and structural features are key for classification accuracy.** 5000 random 5'UTR examples and 5000 random RS examples were used to compute feature importances by scrambling each feature randomly 10 times and calculating the accuracy loss. Features were scrambled one at a time. This calculation was done using sklearn's permutation_importance function. Box plots of the accuracy loss across the entire ensemble were constructed with

the entire output results (10 runs by 20 classifiers by 74 features). Individual dots represent the average accuracy loss of one of the twenty classifiers across its 10 scrambled runs. Mean Free Energy (MFE) and Unbranched stacks (UBS) as well as GC% had a marked decrease in ensemble performance when scrambled; Structural features also tended to be considered more important that most of the sequence features. Certain sequence triplets such as CCG, CGA, GGU, and CUC also were shown to be important nucleotide triplets to our ensemble.

**S3 Fig. Ensemble training with exon and random sequences broken down by classifier.** A secondary ensemble was trained using presumed riboswitch negative sequences (random and exon). The false positive rate for each of the 20 classifiers is shown for two output selection thresholds, 0.5 and 0.95.

## Computation

All processing was done in Python 3.8 with Biopython [68], NumPy [69], and NUPACK 4.0.0.23. Final data was stored in .csv, .npy, and .json files and large files can be recomputed by the reader with data extraction files. All data and analyses files are available at https://github.com/MunskyGroup/human_riboswitch_hits. A modified BEAR encoding in Python was used for structural feature counting from dot structure strings [70] in the file rs_functions.py. All figures can be recreated using the file run_analyses.py.

## Positive unlabeled machine learning

Unweighted Elkan & Noto classifiers were used for our machine learning classifiers. In brief, this is an extension of a generalized probability classifier to train on unknown / known labels as an approximation of class labels. Each data point $x$ has a label $y$ which is either $0$ or $1$. Along with the label pair each data point has a known or unknown flag, $s$, where $s = 1$ if the data point is known, and $s = 0$ when unknown. Therefore, when $s = 1$, $y = 1$ and when $s = 0$, $y = \{0, 1\}$ Any binary classifier is then used to estimate $p(s = 1|y = 1, x)$ instead of the classical estimate of $p(y, x)$. For our paper we used a SVC classifier from sci-kit learn with the following options: SVC(C=10, kernel=rbf, gamma=0.4, probability=True). All PU classifiers were made using an implementation from the Python package PUlearn. For full details refer to the original paper by Elkan & Noto [27].

## GO analysis

GO analysis was performed with the PANTHER overrepresentation test (release 10.13.2022) using the 07.01.2022 release of the PANTHER database (10.5281/zenodo.6799722Released 2022-07-01) using the Fisher's Exact test with False Discovery Rate correction. The reference list used for comparison analysis was the *Homo Sapiens* gene list. Overrepresentation test was performed for the GO biological process complete and GO biological function complete annotated data sets.

## Acknowledgments

Special thanks to Dr. Hamid Chitsaz as this paper started as a student project in his CS548 - Bioinformatics class. Additional special thanks to Dr. Jeffrey Wilusz as the initial idea for this was broached as a class discussion in MIP 543 - RNA biology.

## Author contributions

**Conceptualization:** William Raymond, Jacob DeRoo.

**Data curation:** William Raymond, Jacob DeRoo.

**Formal analysis:** William Raymond.

**Funding acquisition:** Brian Munsky.

**Investigation:** William Raymond.

**Methodology:** William Raymond.

**Project administration:** William Raymond.

**Resources:** Brian Munsky.

**Software:** William Raymond.

**Supervision:** William Raymond, Brian Munsky.

**Validation:** William Raymond.

**Visualization:** William Raymond.

**Writing – original draft:** William Raymond, Jacob DeRoo, Brian Munsky.

**Writing – review & editing:** William Raymond, Jacob DeRoo, Brian Munsky.

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
