## [Decision Letter · Decision Letter 0]

2 Feb 2025

PONE-D-25-01707Identification of potential riboswitch elements in *Homo Sapiens* mRNA 5'UTR sequences using Positive-Unlabeled machine learningPLOS ONE

Dear Dr. Raymond,

Thank you for submitting your manuscript to PLOS ONE. We are pleased to inform you that your manuscript has been provisionally accepted for publication in PLOS ONE, pending the incorporation of revisions based on the enclosed comments from Reviewer 2. Therefore, we invite you to submit a revised version of the manuscript that addresses the points raised during the review process. 

We look forward to receiving your revised manuscript.

Kind regards,

Sudarshan Kasireddy, Ph.D.

Academic Editor

PLOS ONE

Journal Requirements:

WSR, and BM were supported by the NSF (1941870) and National Institutes of Health (R35GM124747). JD was supported by the National Institutes of Health (1R01AI168459-01A1). Special thanks to Dr. Hamid Chitsaz as this paper started as a student project in his CS548 - Bioinformatics class. Additional special thanks to Dr. Jeffrey Wilusz as the initial idea for this was broached as a class discussion in MIP 543 - RNA biology. 

WR and BM were supported by the National Science Foundation (NSF) (1941870) and the National Institutes of Health (NIH) (R35GM124747). The funders played no specific role in design, data collection, analyses or decision to publish for this manuscript.

Reviewers' comments:

Reviewer's Responses to Questions

**Comments to the Author**

1. Is the manuscript technically sound, and do the data support the conclusions?

Reviewer #1: Yes

Reviewer #2: Yes

2. Has the statistical analysis been performed appropriately and rigorously? 

Reviewer #1: N/A

Reviewer #2: Yes

3. Have the authors made all data underlying the findings in their manuscript fully available?

Reviewer #1: Yes

Reviewer #2: Yes

4. Is the manuscript presented in an intelligible fashion and written in standard English?

Reviewer #1: Yes

Reviewer #2: Yes

5. Review Comments to the Author

Reviewer #1: The authors have thoroughly addressed the comments raised by the previous reviewers, demonstrating careful attention to detail and a commitment to improving the manuscript. In my opinion, the article, in its current form, meets the standards of quality and scientific rigor expected for publication in PLOS ONE journal.

Reviewer #2: The authors have satisfactorily addressed the previous reviewers comments. I have one additional comment to the authors.

• This study on identifying potential riboswitch elements in human mRNA 5'UTR sequences provides valuable insights. To enhance the context of the findings, I recommend citing the article "Regulation of mRNA translation by a photoriboswitch". (https://elifesciences.org/articles/51737) This work discusses how riboswitches, including photoriboswitches, regulate mRNA translation in response to light, offering valuable insights into the functional roles of riboswitches. Citing this paper would further highlight the potential applications of the riboswitch elements identified by the authors, particularly in controlling gene expression.

6. PLOS authors have the option to publish the peer review history of their article (what does this mean?). If published, this will include your full peer review and any attached files.

Reviewer #1: No

Reviewer #2: **Yes: **Sri Hari Galla

---

## [Author Response · Author response to Decision Letter 1]

14 Feb 2025

Dear PLOS ONE editors and manuscript reviewers,

We would like to express our gratitude for the editors and reviewers' time. We have endeavored to address all the comments raised about formatting from the editors and comments on content from the reviewers as well as some small general corrections. Below is our response to each numbered point in the decision letter.

Response to editors

The authors have checked the formatting style guidelines for PLOS ONEand ensured that their manuscript adheres to these requirements.

We have checked that our code meets the requirements of PLOS ONE’s open source publishing.

We have removed the following funding statement from the manuscript: “WSR, and BM were supported by the NSF (1941870) and National Institutes of Health (R35GM124747). JD was supported by the National Institutes of Health (1R01AI168459-01A1).” We have moved this statement to the cover letter as requested.

We have reviewed author affiliations and made no changes

The authors have reviewed our cited references, one reference (22, riboflow) was fixed from referencing a preprint to its new publishing location. Several DOI links were corrected as they had extraneous links to specific parts of manuscripts. Now these refer to their manuscript.

Reviewer comments

> Reviewer #2: The authors have satisfactorily addressed the previous reviewers comments. I have one additional comment to the authors.

• This study on identifying potential riboswitch elements in human mRNA 5'UTR sequences provides valuable insights. To enhance the context of the findings, I recommend citing the article "Regulation of mRNA translation by a photoriboswitch". (https://elifesciences.org/articles/51737) This work discusses how riboswitches, including photoriboswitches, regulate mRNA translation in response to light, offering valuable insights into the functional roles of riboswitches. Citing this paper would further highlight the potential applications of the riboswitch elements identified by the authors, particularly in controlling gene expression.

The authors agree that this paper has merit, and synthetic riboswitch uses in synthetic biology should be mentioned in the manuscript. We have added the following sentence to the introduction to address this comment:

“Besides being a critical regulatory tool in nature, both natural and synthetic riboswitches have been used in synthetic biology for responsive translational control, splicing control, and photo-regulation~\cite{Etzel2017,Kelvin2023,Rotstan2020}”

Additional changes

We have fixed an error in the introduction where we say our ensemble’s accuracy on withheld synthetic riboswitches was 56%. This does not reflect the actual section in results where we report an 84% accuracy. This discrepancy between sections is due to a previous coding bug. An earlier version of the manuscript reported an accuracy of 56% due to one structural feature input being incorrectly normalized. When this was fixed, classification jumped from 56% to 84% at a 0.95 threshold and we neglected to correct the introduction.

Additionally, we have changed the following sentence: “...correctly discovered as riboswitches despite having no representation of similar synthetic riboswitches in any training data.” to “...correctly discovered as riboswitches despite having minimal representation of similar synthetic riboswitches in any training data.” This change is to reflect that a tiny percent (14/~60k) of synthetic switches are included in the training data, even though they aren’t the 25 theophylline switches used for validation.

Please let us know if there are any additional requirements from us, and we would like to thank the reviewers and editors again for their time.

Sincerely,

Dr. William Raymond

School of Biomedical Engineering

Colorado State University

---

## [Editor Report · Decision Letter 1]

18 Feb 2025

Identification of potential riboswitch elements in *Homo Sapiens* mRNA 5'UTR sequences using Positive-Unlabeled machine learning

PONE-D-25-01707R1

Dear Dr. Raymond,

We’re pleased to inform you that your manuscript has been judged scientifically suitable for publication and will be formally accepted for publication once it meets all outstanding technical requirements.

Kind regards,

Sudarshan Kasireddy, Ph.D.

Academic Editor

PLOS ONE

---

## [Editor Report · Acceptance letter]

PONE-D-25-01707R1

PLOS ONE

Dear Dr. Raymond,

I'm pleased to inform you that your manuscript has been deemed suitable for publication in PLOS ONE. Congratulations! Your manuscript is now being handed over to our production team.

Kind regards,

on behalf of

Dr. Sudarshan Kasireddy

Academic Editor

PLOS ONE